# Association of Genomic Instability Score, Tumor Mutational Burden, and Tumor-Infiltrating Lymphocytes as Biomarkers in Uterine Serous Carcinoma

**DOI:** 10.3390/cancers15020528

**Published:** 2023-01-15

**Authors:** Elizabeth A. Bloom, Pamela N. Peters, Regina Whitaker, Shonagh Russell, Benjamin Albright, Shelly Cummings, Kirsten M. Timms, Thomas Slavin, Braden Probst, Kyle C. Strickland, Rebecca A. Previs

**Affiliations:** 1Duke University School of Medicine, Duke University, Durham, NC 27710, USA; 2Department of Obstetrics and Gynecology, Division of Gynecologic Oncology, Duke University School of Medicine, Durham, NC 27710, USA; 3Department of Pharmacology and Cancer Biology, Duke University School of Medicine, Durham, NC 27710, USA; 4Myriad Genetics Laboratories, Inc., Salt Lake City, UT 84108, USA; 5Department of Pathology, Duke University, Durham, NC 27708, USA; 6Labcorp Oncology, Durham, NC 27560, USA

**Keywords:** uterine serous cancer, tumor mutational burden, genomic instability, tumor-infiltrating lymphocytes, immunohistochemistry

## Abstract

**Simple Summary:**

Uterine serous carcinomas (USC) represent a rare and aggressive subtype of uterine cancer. Patients often receive adjuvant therapy including chemotherapy and/or radiation after surgery, which has limited efficacy in preventing high rates of recurrence. Understanding the tumor microenvironment in USC is paramount to developing new targeted therapies at the time of progression. This study evaluated the genomic instability score (GIS), tumor mutational burden (TMB), and tumor-infiltrating lymphocytes (TILs) in 53 patients with USC. In this cohort, the median TMB was 1.35 mutations/megabase (mt/Mb); patients with TMB greater than the median had improved survival outcomes. The median GIS was 31, and a higher GIS was not associated with improved survival. We characterized immune cell populations and found that increased immune populations were not associated with a better prognosis.

**Abstract:**

Background: Uterine serous carcinomas represent 10% of uterine carcinomas but account for nearly 40% of deaths from the disease. Improved molecular characterization of these tumors is instrumental in guiding targeted treatment and improving outcomes. This study assessed the genomic instability score (GIS), tumor mutational burden (TMB), and tumor-infiltrating lymphocytes (TILs) in patients with USC. Methods: A retrospective cohort study evaluated patients with USC following staging surgery. The GIS and TMB were determined from archived specimens. We evaluated the tumoral expression of CD3, CD4, CD8, FOXP3, and CD68 using immunohistochemistry. T-tests were used to assess associations of TILs with the GIS. Results: We evaluated 53 patients with USC. The median GIS was 31 (range: 0–52) and a higher GIS was not associated with progression-free (PFS) or overall survival (OS). The median TMB was 1.35 mt/Mb; patients with TMB > 1.35 mt/Mb had improved PFS and OS (*p* = 0.005; *p* = 0.002, respectively). Tumors with increased CD3+ and CD4+ immune cells had a higher mean GIS (*p* = 0.013, *p* = 0.002). Conclusions: TMB > 1.35 mt/Mb was associated with improved survival in USC patients, whereas the GIS was not. Lower TMB thresholds may provide prognostic value for less immunogenic tumors such as USC. In this limited cohort, we observed that increased TIL populations were correlated with a higher GIS.

## 1. Introduction

### 1.1. Biomarker-Directed Care in Uterine Serous Cancer

Uterine serous cancer is a rare, aggressive histological subtype of uterine cancer, the most common gynecological malignancy in the United States, and comprises approximately 10% of all uterine cancer cases [1,2]. However, it accounts for 40% of deaths from uterine cancer [3]. Due to its aggressive behavior, patients with uterine serous cancer often present at an advanced stage (38%) compared to patients who have the more common endometrioid histology (16%) [4]. Those who present with advanced-stage disease have recurrence rates as high as 90% [4]. 

In the era of precision medicine, biomarker-targeted therapies are increasingly incorporated into treatment algorithms. Patients with uterine serous cancer represent a population with a high unmet clinical need. The current standard of care for patients with uterine serous cancers includes staging surgery followed by adjuvant therapy including chemotherapy and/or radiation [5]. Recently, checkpoint inhibitors have been approved for patients with recurrent uterine cancer depending on whether a tumor is microsatellite stable (MSS) or unstable (MSI-H). Single-agent pembrolizumab and dostarlimab, both PD-1 inhibitors, are approved for MSI-H and mismatch deficient tumors in the recurrent or metastatic setting [6,7]. Pembrolizumab is also approved in combination with lenvatinib, a multiple receptor tyrosine kinase inhibitor, for patients with recurrent MSS uterine cancer, including those with serous histology [8]. The incorporation of trastuzumab to treat uterine serous cancer with *HER2* overexpression has improved survival for patients with advanced and recurrent disease [9]. Evaluating additional biomarkers for incorporation into precision medicine will be essential to improve outcomes for patients with uterine serous cancer. 

### 1.2. Tumor-Infiltrating Lymphocytes (TILs)

In several solid malignancies, including lung and epithelial ovarian cancers, higher densities of tumor-infiltrating lymphocytes (TILs) correlate with improved survival and response to immune checkpoint blockade [10,11,12,13]. The role of TILs as a biomarker for the response to immune checkpoint blockade is unclear in all uterine cancers, including the serous subtype. It is generally believed that high CD8+ T-cells are associated with better prognosis and that tumor-associated macrophages (TAMs) and regulatory T-cells (FOXP3+) are associated with worse survival in uterine cancer, but study conclusions vary [14,15,16,17]. These studies primarily included patients with the endometrioid subtype of uterine cancer. Only in recent years has attention begun to shift toward the uterine serous subtype specifically; Mise et al. highlighted that CD8+ cells were associated with improved prognosis in a cohort of 62 patients with USC. Additionally, these recent studies have looked at other immune signals, such as CCL7 and PLK3, and their effects on the immune microenvironment of uterine serous cancer [18,19]. Therefore, there is an unmet need for studies that stratify patients based on histologies to assess differences in TILs and outcomes for patients with uterine serous tumors.

### 1.3. Tumor Mutation Burden (TMB)

Resistance to checkpoint inhibitors in uterine serous cancer has been attributed to an immunosuppressive tumor microenvironment or a “cold tumor”. Cold tumors have a low diversity of tumor neo-antigens, few coding mutations, and dysregulation of T-cell trafficking to the immune microenvironment [8,20]. Thus, for immune checkpoint inhibitors to be effective in patients with uterine serous cancer, appropriate adjunct therapies are required to bolster the immune response by turning the “cold tumor” into a “hot tumor”, thereby increasing immunogenicity and response to checkpoint blockade. TMB may represent a surrogate marker for tumor immunogenicity and has been evaluated as a biomarker of response to checkpoint inhibition. In the clinic, TMB ≥ 10 mt/Mb is used as a biomarker for single-agent pembrolizumab in solid tumors [21,22]. However, the relationship between TMB and outcomes in patients with uterine serous carcinoma is not well established. Although the Food and Drug Administration approval for pembrolizumab included patients with uterine cancer, few uterine cancers meet this criterion for TMB-high, as the average TMB for uterine cancers ranges from 2.6–4.5 when accounting for all histologic subtypes [23,24]. As such, evaluating different TMB thresholds for uterine serous cancer may enable the successful utilization of immunotherapies in these patients.

### 1.4. Homologous Recombination Deficiency (HRD) and Molecular Characterization

An additional biomarker with therapeutic and prognostic significance in gynecologic malignancies is homologous recombination deficiency (HRD). HRD genes include *BRCA1*, *BRCA2, RAD51C, RAD51D, BARD1, BRIP1,* and *PALB2.* Although the site of origin differs, approximately 50% of patients with high-grade serous carcinomas of the ovary (HGSOC) have underlying HRD [25], and they also share histologic and genotypic similarities to uterine serous cancers [26]. Additionally, patients with HGSOC whose tumors exhibit HRD have been found to have a higher density of TILs. These patients with both HRD and higher TILs density had improved outcomes [27]. Next-generation sequencing (NGS) allows for the comprehensive evaluation of genetic biomarkers, including the HRD genes. NGS also allows for the evaluation of those genes required for molecular classification, such as the mismatch repair genes, *MLH1, MSH2, MSH6*, and *PMS2,* as well as *TP53, ERBB2,* and *POLE* [25,26]. Given the genotypic similarities of HGSOC and uterine serous cancers, we aimed to assess underlying rates of HRD in uterine serous cancers, as defined by a genome instability score (GIS), in uterine serous cancers and correlate these scores with immune cell infiltration. 

### 1.5. Objective and Plan

The objective of this study was to evaluate the association between HRD, TMB, and immune cell infiltration in a cohort of patients with uterine serous cancer to determine the contribution of each biomarker to clinical outcomes. 

## 2. Materials and Methods

### 2.1. Patient Population

A single institution, retrospective cohort study was performed on patients diagnosed with high-grade serous uterine cancer following hysterectomy and staging surgery at Duke University Medical Center following Institutional Review Board Approval (Pro00100761, Pro00008216). A board-certified pathologist (KS) reviewed each case to confirm diagnosis. Inclusion criteria were patients ≥18 years old with a diagnosis of uterine serous cancer and who underwent hysterectomy at Duke University Hospital between 1 January 2008 and 5 January 2018. Patients were excluded if they had received treatment for uterine cancer prior to surgery. 

### 2.2. Genetic Tumor Testing

Archived tumor samples were submitted for molecular testing by Myriad Genetics. Analysis included NGS testing for somatic variants in tumor samples. A Genomic Instability Score (GIS) was determined for each case. GIS is an algorithmic measurement of Loss of Heterozygosity (LOH), Telomeric Allelic Imbalance (TAI), and Large-scale State Transitions (LST) using DNA isolated from formalin-fixed paraffin-embedded (FFPE) tumor tissue specimens. TMB was assessed using a SNP resequencing assay as described by Timms et al. [28]. 

### 2.3. Immunohistochemistry

Formalin-fixed paraffin-embedded (FFPE) tissues were evaluated via immunohistochemistry (IHC) for expression of CD3, CD4, CD8, FOXP3, and CD68. Sections were deparaffinized in xylene and declining grades of ethanol prior to rehydration. After antigen retrieval with citrate buffer pH 6.0 (Sigma-Aldrich, St. Louis, MO, USA), the sections were blocked with 3% hydrogen peroxide and protein blocking solution (Background Terminator, BioCare Medical, Pacheco, CA, USA) at room temperature. The sections were incubated with the following primary antibodies overnight at 4 °C: CD3 (1:250 dilution, clone F7.2.38, Dako, Santa Clara, CA, USA), CD4 (ready to use [RTU] dilution, clone 4B12, Leica, Buffalo Grove, IL, USA), CD8 (RTU dilution, clone 4B11, Leica, Buffalo Grove, IL, USA), FOXP3 (RTU, clone 236A/E7, Leica, Buffalo Grove, IL, USA), and CD68 (1:2 of RTU, clone PG-M1, Dako, Santa Clara, CA, USA). After washing the slides with tris-buffered saline, protein expression was visualized using the 4 Plus Universal Detection system (BioCare Medical, Pacheco, CA, USA) for 10 min at room temperature. Slides were developed with 3,3”-diaminobenzidine chromogen (Vector Laboratories, Burlingsame, CA, USA) and counterstained using a modified Lillie-Mayer Hematoxylin (BioCare Medical, Pacheco, CA, USA).

A board-certified anatomical pathologist (KS) evaluated immune cell immunohistochemistry as previously described [29]. Briefly, a photomicrograph of the area of maximum CD3+ intraepithelial lymphocytes was obtained (40× objective), and photomicrographs from the corresponding area were obtained for the additional stains. Counts were performed manually in Photoshop. For evaluation of TILs, we focused only on intraepithelial lymphocytes defined as lymphocytes located within the tumor epithelium, rather than in the peritumoral stroma. The number of CD3+ intraepithelial lymphocytes was manually counted in one high-power field (40× objective) of the highest density of TILs. Protein expression in adjacent normal endometrial glands and stroma served as internal controls. 

We evaluated expression for CD3, CD4, CD8, FOXP3, and CD-68 for 42 tumors. Eleven cases were excluded because of insufficient tumor identified on the slides or inability to obtain pathology samples. We evaluated the distribution of immune cell counts by quartiles to identify those tumors with the highest immune infiltrate. A high (H) cell count was defined as greater than the 75th percentile. A low–normal (L–N) count was defined as less than or equal to the 75th percentile. 

### 2.4. Clinical Data Collection

Clinical and demographic data were abstracted from the electronic medical record. The abstracted data included age, race, ethnicity, body mass index, parity, performance status, medical co-morbidities, surgical procedure, stage, adjuvant treatment, recurrence status, and vital status. 

### 2.5. Statistics

Descriptive statistics were used to describe the clinical and demographic characteristics of the population studied. Chi-squared tests were used to evaluate differences in these groups. Univariate and multivariate Cox proportional hazards models were used to determine associations of clinical and molecular factors with progression-free survival (PFS) and overall survival (OS), with age, stage, and ECOG performance status used as covariates. Chi-squared tests were used to evaluate differences in TILs infiltration based on TMB and GIS, as well as rates of recurrence between subgroups of tumors. T-tests were used to evaluate differences in mean TMB and GIS between tumor groups, with significance level of *p* < 0.5.

## 3. Results

### 3.1. Patient Population

A total of 53 patients were included in the cohort; the median age of presentation was 68.1 years (range 55.3–88.7). The majority of patients presented with advanced disease; 26.4% were diagnosed at Stage IV (*n* = 14/53), 39.6% at Stage III (*n* = 21/53), and 33.9% (18/53) at Stage I or II. At the time of analysis, 71.7% of patients (*n* = 38/53) had recurred, and 62% (*n* = 33/53) had died of recurrent disease. The median PFS for this population was 21.7 months, and the median OS was 33.6 months. The GIS and TMB were determined for each patient. The GIS was obtained for 41 of 53 patients, with a median of 31 (range 0–52). TMB was obtained for 52 of 53 patients with a median of 1.35 mt/Mb (range 0–15.8) (Table 1). 

### 3.2. Genomic Landscape of Uterine Serous Carcinoma

Next-generation sequencing (NGS) was obtained for all tumor samples to identify somatic mutations. *TP53* was the most commonly identified pathogenic mutation (37/53 tumors, 69.8%). *POLE* mutations were the second most common pathogenic mutation (12/53 tumors, 22%), but these mutations had unknown pathogenic significance (Figure 1). Of these 12 patients, 8 (67%) had a recurrence of the disease. No tumors had a likely pathogenic mutation involving the HRD pathway; 1 tumor had a pathogenic mutation in *MSH6*, part of the mismatch repair (MMR) pathway (Figure 2, Appendix A). This patient was the only patient identified with microsatellite instability (Table 1, Appendix A). Mutations were identified in other HRD genes, but their pathogenic significance was unknown. The specific mutations identified in the HRD pathway are delineated by HRD status (Appendix A). TMB was obtained for 52 patients. The median TMB was 1.35 mt/Mb, and the mean was 2.26 mt/Mb (range: 0–15.4). 

### 3.3. Tumor Recurrence and Genomic Instability among Immune Cell Subgroups

To characterize immune populations within the tumor microenvironment, we evaluated the expression of CD3, CD4, CD8, FOXP3, and CD68. Representative photomicrographs of tumors with high and low–normal immune cell infiltration are shown (Figure 2).

We evaluated the distribution of uterine serous tumors in quartiles defined by GIS and immune cell infiltration. Across all immune cell types, less than 10% of tumors had both elevated immune cell counts and low GIS scores. When evaluating CD3+ immune cell populations specifically, only 2 (5%) tumors had a low GIS and high immune cell infiltration (*p* = 0.126). No patients had tumors with high populations of CD4+ immune cells and a low GIS (*p* = 0.002). Three tumors (7%) had an increased CD8+ immune cell infiltration and a low GIS (*p* = 0.446). Four tumors (10%) had increased FOXP3+ immune cells and a low GIS (*p* = 0.871). Two tumors (5%) had high CD68+ immune cells and a low GIS (*p* = 0.394). Similarly, few tumors (≤15%) had both elevated immune cell counts and TMB below the median (Table 2).

Thirty-eight patients (71.7%) developed recurrent disease; 37 patients (69%) of the entire cohort had sufficient tumor for evaluation by both IHC and NGS. We observed fewer recurrences in patients whose tumors had a lower TIL infiltration and a GIS below the median. For example, 17 patients had tumors with a GIS below the median and low–normal CD3+ infiltration, and 53% (9/17) of those developed recurrence. By contrast, 7 patients had tumors with an elevated GIS and high CD3+ TIL infiltration and 86% (6/7) of these patients developed recurrence. (Figure 3A–C) (*p* = 0.20). When evaluating patients by TMB and TILs, there were no differences in recurrence rates (Figure 3D–F).

We compared the mean GIS for tumors with high and low–normal immune cell populations. Tumors with higher infiltration of CD3+ immune cells had a significantly higher mean GIS than those with low–normal CD3+ lymphocytes (38.7 vs. 27.9, *p* = 0.013 (Figure 4A)). This trend was also observed for tumors with high CD4+ immune cell populations, and a significantly higher mean GIS was observed than for those with low–normal CD4+ immune cell populations (40.1 vs. 27.3, *p* = 0.003) (Figure 4B). There was no significant difference between mean GIS scores and CD8+ immune cell populations (Figure 4C) (*p* = 0.2), or CD68+ and FOXP3+ immune cells (data not shown). There was also no significant difference in the mean TMB for tumors with high and low–normal immune populations (Appendix A).

### 3.4. Survival Analysis

On univariate analysis, TMB > 1.35 mt/Mb was associated with improved PFS and OS (*p* = 0.005 and 0.0019) (Figure 5A,B). On Cox multivariate regression analysis, when using age, stage, and ECOG performance status as covariates, TMB > 1.35 mt/Mb continued to be associated with improved PFS and OS (*p* = 0.031 and 0.000) (Table 3). The GIS was not associated with PFS or OS (Figure 5C,D). We also evaluated survival based on immune cell populations and no significant differences in outcomes were identified. 

## 4. Discussion

This retrospective cohort study described the genomic instability score, tumor mutational burden, tumor-infiltrating immune cells, and clinical outcomes for patients with uterine serous cancer. In this population of 53 patients with uterine serous carcinoma, the median TMB was 1.35 mt/Mb and the median GIS was 31. The majority of patients (70%) had tumors with pathogenic *TP53* mutations. No patients had pathogenic mutations in the HRD pathway, and one had a mutation in the mismatch repair pathway (*MSH6*). In our cohort of patients with uterine serous cancer, the GIS was not associated with clinical outcomes including PFS or OS. Patients with TMB greater than the median of 1.35 mt/Mb had improved PFS and OS, as compared to those with TMB ≤ 1.35 mt/Mb. Immune cell populations were quantified for CD3+, CD4+, CD8+, FOXP3+, and CD68+ cells. A higher average GIS score was observed in tumors with a high density of CD3+ and CD4+ immune cells. Fewer recurrences were observed in patients whose tumors had fewer TILs and a GIS below the median. 

Although uterine serous cancers bear a histologic resemblance to HGSOC, this study provides early evidence that these similarities may not extend into the prognostic value of genomic and immunologic biomarkers. In high-grade epithelial ovarian cancers, genomic instability was associated with a higher immune cell infiltration and improved survival; however, in our cohort, we observed the opposite trend [27]. This difference may be due to an alternate role of the immune system in uterine serous cancers or unique conditions in the tumor microenvironment. Further studies in an expanded cohort may improve the understanding of this trend observed in patients.

Prior research on TILs in uterine cancer has focused primarily on the endometrioid subtypes; however, this study focused on the rare, less-studied subtype of uterine cancer to better understand the immune microenvironment and clinical implications [15,16,17]. Only a few select studies to date have looked at the immune microenvironment, and have focused on molecular signals and markers such as CCL7 and PLK3 [18,19]. Guo et al., in a meta-analysis, showed that patients with a higher CD8+ immune cell infiltration trended towards improved survival, but this was not stratified based on uterine histological subtypes. Mise et al. showed improved survival in those with an elevated CD*+ infiltration, but this trend was not seen in our limited cohort in this study. The meta-analysis also looked at studies that examined FOXP3+ immune cells and TAMs, and found no correlation with survival [17]. Previous studies failed to define the relationship between immune cell populations and patient survival in uterine cancer as a whole. Additional studies have utilized immune cell infiltration to categorize the tumor microenvironment into immune subtypes, such as IFN-γ-dominant, inflammatory type, wound healing type, and immunologically balanced type. Li and Wan showed that certain subtypes with the highest immune cell infiltration had the best survival outcomes [30]. This study showed that the subtype with dominant IFN-γ and low macrophage infiltration had the worst survival outcomes. While this study did not identify differences in outcomes based on immune cell populations, further analysis of a larger uterine serous cancer population is warranted to determine the significance of immune cell infiltration in these patients’ overall survival.

Although this study did not show the prognostic significance of the GIS in USC, clinical trials are ongoing in ovarian cancer to investigate the value of the GIS as a biomarker for using poly-adenosine diphosphate-ribose polymerase (PARP)-inhibitors. These studies highlight the prognostic significance of the GIS when selecting PARP inhibitor treatment or maintenance, but consensus on the GIS score that defines elevated HRD is lacking [31,32]. In most of these trials, the cutoff for elevated HRD in high-grade serous ovarian cancers is a GIS of 42. In one trial (NCT02470585), a cutoff of 33 was used [33]. Other studies have recommended a cutoff for GIS as low as 4 for endometrial cancer [34]. The median GIS in this cohort of serous uterine cancers was 31, and 7 patients (17.1%) had a GIS ≥ 42. If this trend persists in larger cohorts of patients with USC and survival differences are observed, an alternate threshold of the GIS may help to identify patients whose tumors could respond to PARP inhibition. While HRD status is currently used as a discrete variable, with patients treated as either positive or negative, the utility of different cutoffs in different cancers raises the possibility of treating the GIS as a continuous variable instead. Trials of PARP inhibitors are ongoing and enrolling patients with uterine serous cancer, so it will be important to support further translational work to understand the GIS landscape in these tumors and reach a consensus about utilizing this score [35].

Tumors with elevated GIS are generally considered to be more immunogenic [36]. More genomic instability and variation generate more neoantigens and alterations to the tumor microenvironment that may stimulate an immune response. This study observed a significantly higher mean GIS in tumors with higher CD3+ and CD4 + immune cells. While this is a preliminary observation, these findings may suggest that the presence of these immune cells may indicate the underlying GIS. This trend was not as prominent for CD8+, FOXP3+, or CD68+ immune cells in this cohort of uterine serous carcinomas. This association of immune cells with the GIS is consistent with the finding in HGSOC and that HRD is associated with a higher TIL infiltration [27]. A recent study by Lea et al. in colorectal cancer also classified tumors by levels of immune cell infiltration. They found that the tumors with high CD3+ and CD8+ immune cells tended to have earlier-stage disease and higher rates of microsatellite instability [37]. This, in combination with our preliminary finding, suggests that further investigation into the correlation of TILs with other immune markers is warranted. 

This study has emphasized that biomarker cutoffs cannot be universally applied across cancer types, both for the GIS as well as TMB. The FDA approved tumor agnostic use of pembrolizumab for patients whose tumors have TMB ≥ 10 mutations/Mb, but few uterine serous cancers meet this criterion. In this patient cohort, only one tumor had a TMB above 10. Patients with recurrent uterine cancers are eligible for checkpoint blockade in the recurrent setting regardless of TMB. However, TMB may represent a prognostic biomarker in uterine cancer. Our work showed that a modest elevation in TMB (defined as >1.35 mt/Mb) was associated with an improved prognosis. This is consistent with literature showing that TMB is a prognostic marker in many other cancers, including breast and colorectal [24,38,39]. One study showed that those with elevated TMB, defined as TMB in the top 20% within each cancer type, had better overall survival, even though TMB distributions varied widely across cancer types [39]. In this study, the cutoff for the top 20% ranged from 5.9 mt/Mb in renal cell carcinoma to 52.2 mt/Mb in colorectal cancer; this cohort did not include uterine cancers. Wu et al. showed that for uterine cancer, TMB ranged from 1.58 to 19.47, with a median of 2.63 mt/Mb. This study defined TMB-high as above the 65th percentile (9.74), and showed a statistically significant decrease in mortality for patients in this group [24]. These studies also highlight that the prognostic value of TMB may be in determining which patients respond best to immunotherapies. Even though all patients with recurrent uterine cancer are eligible for checkpoint blockade, understanding the prognostic value of TMB can further inform patients about their response to therapy. 

We acknowledge the inherent limitations of a single institution cohort study with a small sample size. This small sample size limits the statistical power of this study and the ability to draw conclusions that are statistically significant, particularly in analyses that require subdividing the study population into small cohorts for comparison. As such, the conclusions drawn in this study are exploratory and observational in nature. An additional limitation of this study is the difficulty with histotyping and variability in the diagnosis of uterine serous carcinoma. While additional analysis of surrogate biomarkers may have aided in clarifying the diagnosis for these patients, we chose to rely on pathologic diagnosis, as this has the greatest clinical relevance and greatest consistency across the study. This study is also limited by the difference in rates of *TP53* mutation in this cohort (70% pathogenic) compared to the rate published in The Cancer Genome Atlas (~88%). We recognize that this may be due to inherent uncertainty with histotyping uterine serous cancer. This has been addressed by having each case reviewed by a board-certified pathologist. We were unable to calculate GISs for a subset of the tumor samples. Additionally, the threshold of GIS scores elevation in high-grade serous ovarian cancer, of 42, was developed by studying the distribution of GIS scores in BRCA-mutated tumors, but this was not possible in this cohort because no patients had BRCA-mutated tumors. Our study’s strengths include a pathologist reviewing all cases to confirm pathologic diagnosis and ensuring a relatively homogenous patient population for this rare tumor. 

This study is an observational study to support future work in understanding the immune and molecular profiles of uterine serous cancers. Future prospective studies with a larger number of patients are warranted to clearly establish relationships between these immunologic and molecular factors and patient outcomes. Additional methods of assessing the immune microenvironment will be informative. Conducting flow cytometry in parallel with immunohistochemistry to evaluate TILs infiltration will use fresh tumors and offer a snapshot into the tumor microenvironment. Flow cytometry also lends itself to understanding the proportions of immune cells and allows for the assessment of many subsets of cells simultaneously. Additionally, analyzing TILs in matched primary and recurrent samples in patients who progress on immunotherapy or other targeted therapy may lend insight into how the immune system and tumor biology change and develop resistance to these treatments. Using immunofluorescence to both measure TILs throughout a tumor sample, rather than in a singular slide, and to characterize the interactions of TILs with other cellular markers would provide both a more robust quantification of TILs and information on how these cells interact with the tumor environment.

## 5. Conclusions

In conclusion, this study examined TMB, GIS, and TILs in 53 patients with uterine serous carcinomas. This study demonstrated that an elevated TMB above the median (1.35 mt/Mb) was associated with improved survival outcomes. This study did not find survival differences based on the GIS or TILs. There was a trend toward lower rates of tumor recurrence in patients whose tumors had lower immune cell infiltration and low GIS scores. This exploratory analysis indicates that further research is required to delineate the prognostic significance of TMB in uterine serous cancers. In addition, studies of GIS scores and tumor-infiltrating lymphocytes in a larger cohort of patients with uterine serous cancers are warranted to determine whether the trends observed in this study are consistently observed.

## Figures and Tables

**Figure 1 cancers-15-00528-f001:**
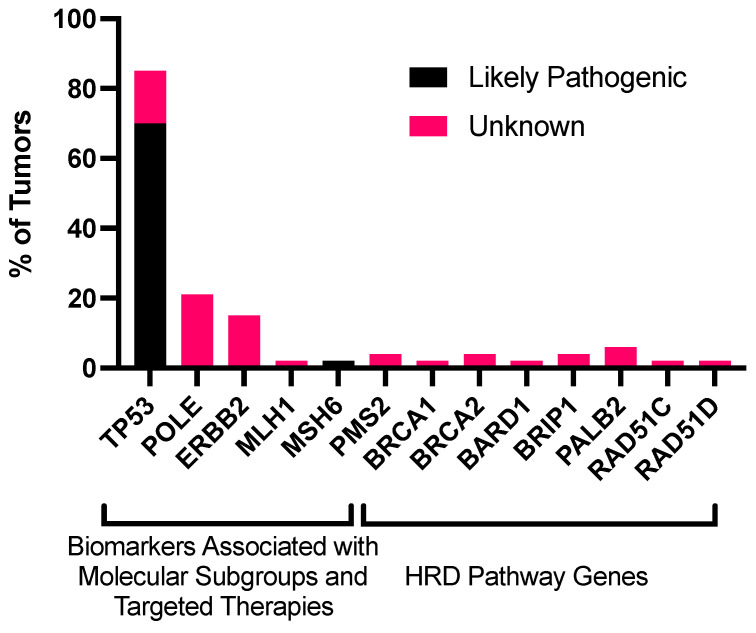
Frequency of mutations in genes associated with molecular classification and other biomarkers (TP53, POLE, ERBB2, and MMR genes) and genes associated with the HRD pathway (BRCA1, BRCA2, PALB2, RAD51C, RAD51D, BRIP1, and BARD1) in uterine serous cancers. This analysis includes point mutations, insertions, deletions, and rearrangements. Unknown mutations were mutations with unknown pathogenic significance. Abbreviations: HRD, homologous recombination deficiency.

**Figure 2 cancers-15-00528-f002:**
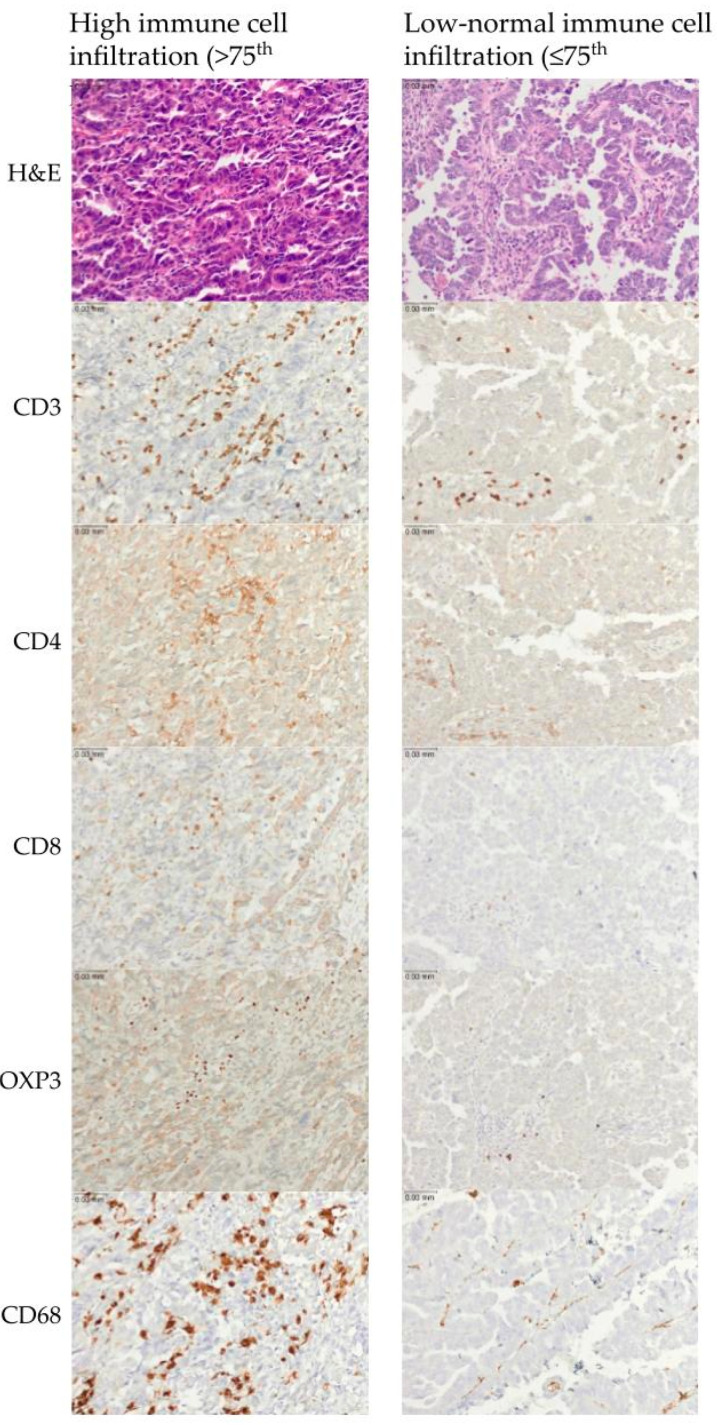
Representative examples of immune cell infiltration for CD3, CD4, CD3, FOXP3, and CD68. The upper row represents high immune cell infiltration (>75th percentile). The lower row represents low–normal immune cell infiltration (≤75th percentile). Scale bar is 0.03 mm.

**Figure 3 cancers-15-00528-f003:**
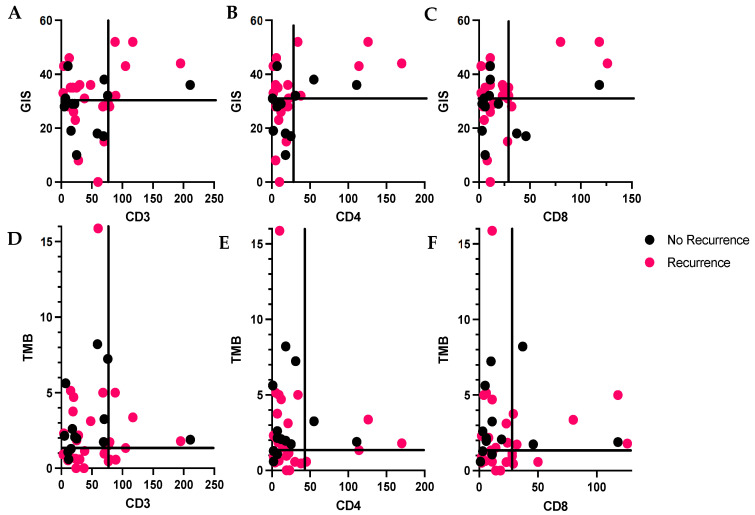
(**A***–***C**): Distribution lymphocytes by genomic instability score (GIS) and tumor recurrence; pink = recurrent, black = no recurrence. Quadrants defined by GIS > 31 (median) or ≤31 and number of TILs > or ≤the 75th percentile. (**D***–***F**): Distribution lymphocytes by tumor mutational burden (TMB) and tumor recurrence. Quadrants defined by TMB > 1.35 (median) or ≤1.35 and number of TILs > or ≤the 75th percentile. (**A**): CD3+ lymphocytes, 75th percentile is 76.8, (**B**): CD4+ lymphocytes, 75th percentile is 28.8, (**C**): CD8+ lymphocytes, 75th percentile is 28.8.

**Figure 4 cancers-15-00528-f004:**
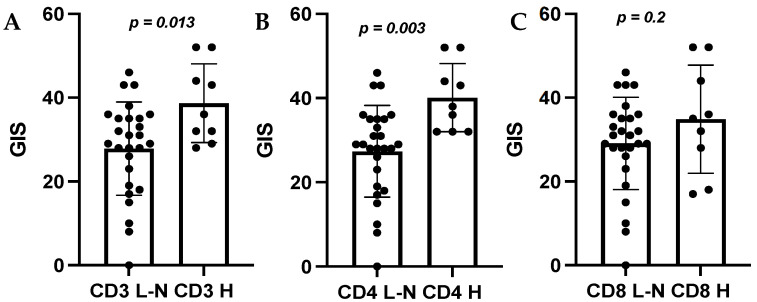
Comparison of mean GIS score by high (H) and low–normal (L–N) immune cell counts. (**A**): CD3, (**B**): CD4, (**C**): CD8. Abbreviations: GIS, genomic instability score.

**Figure 5 cancers-15-00528-f005:**
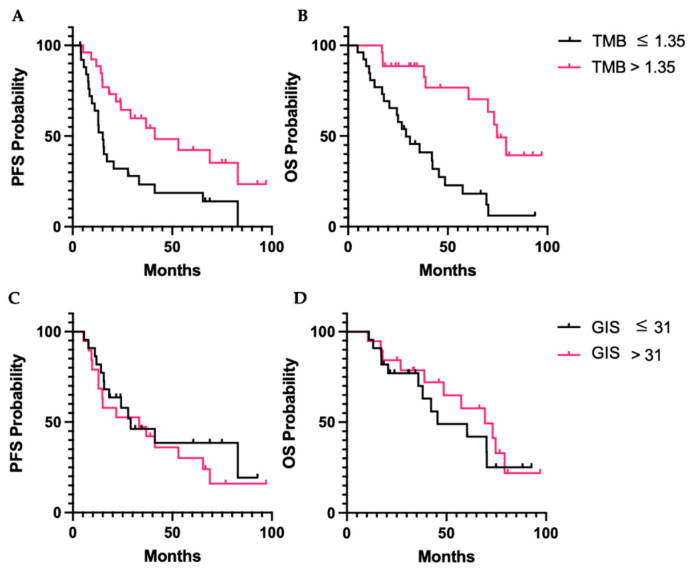
Survival outcomes for patients with uterine serous cancer by TMB and GIS. (**A**,**B**): PFS and OS with TMB above and below or equal to the median, 1.35. (**C**,**D**): PFS and OS for patients with GIS above and below or equal to the median. Abbreviations: GIS, genomic instability score; TMB, tumor mutation burden; PFS, progression-free survival; OS, overall survival.

**Table 1 cancers-15-00528-t001:** Clinical characteristics of patient diagnosed with uterine serous carcinoma. MSS (microsatellite stable), MSI-H (microsatellite instability-high), GIS (genomic instability score), PFS (progression-free survival), OS (overall survival), ECOG (European Cooperative Oncology Group).

Patient Characteristics	All Subjects *n* = 53 (%)	GIS > 31 *n* = 19 (%)	GIS ≤ 31 *n* = 22 (%)	Undetermined *n* = 12 (%)	Χ^2^ (Degrees of Freedom, *n*), *p*
Age at Diagnosis (years)					
50–60	3 (6)	1 (5)	1 (5)	1 (8)	Χ^2^(4, 53) = 2.382, 0.666
60–69	28 (53)	11 (58)	13 (69)	4 (33)	
>70	22 (42)	7 (37)	8 (36)	7 (58)	
Stage					
I	13 (25)	5 (26)	7 (32)	1 (8)	Χ^2^(6, 53) = 22.207, 0.001
II	5 (9)	1 (5)	4 (18)	0 (0)	
III	21 (40)	11 (58)	8 (36)	2 (17)	
IV	14 (26)	2 (11)	3 (14)	9 (75)	
Race					
White	23 (43)	6 (13)	11 (50)	6 (50)	Χ^2^(4, 53) = 4.476, 0.332
Black	28 (53)	11 (58)	11 (50)	6 (50)	
Other	2 (4)	2 (11)	0 (0)	0 (0)	
Ethnicity					Χ^2^(2, 53) = 1.824, 0.402
Not Hispanic or Latino	52 (98)	18 (95)	22 (100)	12 (100)	
Hispanic or Latino	1 (2)	1 (5)	0 (0)	0 (0)	
ECOG Performance Score					
0	16 (30)	3 (16)	6 (27)	7 (58)	Χ^2^(6, 53) = 8.825, 0.184
1	8 (15)	3 (16)	4 (18)	1 (8)	
2	2 (4)	1 (5)	0 (0)	1 (8)	
Unknown	27 (51)	12 (63)	12 (55)	3 (25)	
Surgery Type					
Minimally Invasive	35 (66)	13 (68)	16 (73)	6 (50)	Χ^2^(4, 53) = 3.292, 0.510
Open	16 (30)	5 (26)	5 (23)	6 (50)	
Unknown	2 (4)	1 (5)	1 (5)	0 (0)	
Adjuvant Therapy					Χ^2^(2, 53)
Chemotherapy	45 (85)	17 (89)	18 (82)	10 (83)	0.496, 0.780
Brachytherapy	19 (36)	7 (37)	10 (45)	2 (17)	2.810, 0.245
Whole Pelvic Radiation	21 (40)	9 (47)	9 (41)	3 (25)	1.564, 0.457
Observation	5 (9)	1 (5)	2 (9)	2 (17)	1.125, 0.570
Chemotherapy + Radiation	28 (53)	12 (63)	13 (59)	3 (25)	4.889, 0.087
Microsatellite Status					
MSS	52 (98)	19 (100)	21 (95)	12 (100)	Χ^2^(2, 53) = 14.36, 0.488
MSI-H	1 (2)	0 (0)	1 (5)	0 (0)	
Recurrence and Survival					
Recurrent disease diagnosed	38 (72)	15 (79)	13 (59)	10 (83)	Χ^2^(2, 53) = 3.015, 0.221
Median PFS (months)	21.67 (3.67–97.07)	33.30 (5.33–97.07)	24.03 (5.57–92.73)	14.08 (3.67–82.90)	Χ^2^(2, 53) = 4.654, 0.098
Median OS (months)	33.60 (4.83–97.07)	48.63 (10.53–97.07)	35.17 (11.07–92.73)	26.55 (4.83–93.83)	Χ^2^(2, 53) = 6.285, 0.043

**Table 2 cancers-15-00528-t002:** Quantification of tumor-infiltrating lymphocytes (CD3+, CD4+, CD8+, CD68+, and FOXP3). Counts are stratified by low–normal (L–N, ≤75th percentile) versus high (H, >75th percentile) lymphocyte count and by median GIS and median tumor mutational burden (31 and 1.35, respectively). Abbreviations: GIS, genomic instability score; TMB, tumor mutational burden.

*n* (%)		CD3		CD4		CD8		FOXP3		CD68	
		L-N	H	L-N	H	L-N	H	L-N	H	L-N	H
GIS	GIS > 31	11 (26)	7 (17)	9 (21)	9 (21)	12 (39)	6 (14)	13 (31)	5 (12)	13 (33)	5 (13)
	GIS ≤ 31	17 (40)	2 (5)	19 (45)	0 (0)	16 (38)	3 (7)	15 (36)	4 (10)	16 (41)	2 (5)
	Failed	4 (10)	1 (2)	4 (10)	1 (2)	4 (10)	1 (2)	4 (10)	1 (2)	2 (5)	1 (3)
Χ^2^, *p* dof = 2, *n* = 42		4.144, 0.126		12.784, 0.002		1.614, 0.446		0.276, 0.871		1.861, 0.394	
TMB	TMB > 1.35	18 (44)	6 (15)	18 (44)	6 (15)	16 (39)	8 (20)	16 (39)	8 (20)	18 (46)	6 (15)
	TMB ≤ 1.35	13 (32)	4 (10)	13 (32)	4 (10)	15 (37)	2 (5)	16 (39)	1 (2)	13 (33)	2 (5)
Χ^2^, *p* dof = 1, *n* = 41		0.012, 0.914		0.012, 0.914		2.510, 0.113		4.377, 0.364		0.771, 0.380	

**Table 3 cancers-15-00528-t003:** Progression-free survival and overall survival by TMB: multivariate analysis with age, stage, and ECOG performance status as covariates.

Survival Outcome	Hazard Ratio (95% CI)	*p* Value
Progression-Free Survival	0.419 (0.191–0.924)	0.031
Overall Survival	0.183 (0.741–0.450)	<0.001

## Data Availability

The data presented in this study are available in this article (and Appendix A).

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
