# Peer review of "Association of Genomic Instability Score, Tumor Mutational Burden, and Tumor-Infiltrating Lymphocytes as Biomarkers in Uterine Serous Carcinoma"

_cancers, 2023, doi:10.3390/cancers15020528_

Round 1

Reviewer 1 Report

Comments

1.             Authors need to rewrite the abstract and I would strongly suggest write a simple abstract with graphical representation. The simple summary and abstract background looks same so please remove simple summary. In simple summary (line 26-27) authors mentioned “Median GIS was 31 and higher GIS was not associated with improved survival.” However, in abstract background (line 36-37) “Median GIS was 31 (range: 0-52) and not associated with progression-free (PFS) or overall survival (OS).” These two statements looks contradictory and its very confusing.

2.             Also, I didn’t see any outcome explanation about the immune cells except the methodology in simple summary and abstract background. In graphical abstract authors showed the association of CD3 and CD4 but not CD8. 

3.             In abstract background “Increased TILs populations were correlated with higher GIS “ I am highly disagree with this outcome. Authors need to clarify this. 

4.             Line 78-82; Authors mentioned that CD8+ T-cells are associated with better 78 prognosis and that tumor-associated macrophages (TAMs) and regulatory T-cells 79 (FOXP3+) are associated with worse survival in uterine cancer, but study conclusions vary.. These studies primarily included patients with endometrioid uterine cancer.However, there are some studies available in uterine serous carcinoma where they revealed the precise immune status of USC and  discussed how closely immune microenvironment involved in the malignant traits of cancer.

“Mise, Yuka, Junzo Hamanishi, Takiko Daikoku, Shiro Takamatsu, Taito Miyamoto, Mana Taki, Koji Yamanoi et al. "Immunosuppressive tumor microenvironment in uterine serous carcinoma via CCL7 signal with myeloid-derived suppressor cells." Carcinogenesis (2022).

5.              Line 147-152; I would suggest to please confirm the working dilution of the antibodies (CD4. CD8 and CD68).

6.             Patient data set is very small to get any conclusion. Most patients are above 60 years age and if authors trying to compare different age group than they need to add atleast 10 patient in each group. I think authors need to add more patients samples in their study.

7.             Authors need to improve figure 2. Image quality is not good and scale bar is missing.

8.             I do strongly agree that authors also didn’t find any association between TMB and infiltrating cells. The reports already available which clearly mentioned that In USC, tumor stromal infiltration of CD8+ T cells was found to be not prominent, suggesting that the tumor microenvironment of USC is in a state of antitumor immune tolerance  and this could be reason that existing immune checkpoint inhibitors (ICIs) alone  is not effective to restore antitumor immune status and also because of the low frequency  of highly microsatellite instable (MSI-H) cases and low TMB in USC. Also, it could be tumor (MDSCs) may be  suppress the intertumoral infiltration of CD8+ T cells, and it is reasonably expected that USC employs these mechanisms to hinder the intratumorally infiltration of CD8+ T cells. Therefore as per my understanding, I didn’t find any novelty in the authors manuscript including vey low patient data set.

9.             Whole discussion part is very weak. Authors need to rewrite.

10.          In several instances, the spaces between adjacent words are missing. There are also grammatical and typological mistakes. Please check the spelling and grammar. I would suggest that the whole manuscript be thoroughly revised to improve clarity and readability.

11.          Reference section needs revisions and authors need to cite more recent papers.

Reviewer 2 Report

Bloom et al described GIS, TMB, mution profile and immune infiltration in a series of USC.  The results are interesting, since most studies in these area are focussed on endometrioid carcinomas.

The major limitation of this study is the selection of the cases. Although all cases have been reviewed by an experienced Pathologist, it is well known that histotyping of high-grade endometrial carcinomas is challenging. Even among very experienced Pathologists, disagreement in histotyping is higher than 35% (PMID 23629444). Major disagreement occurred between USC and clear cell carcinoma and between USC and grade 3 endometrioid carcinomas. Agreement increased with the use of IHC.

Taken into account morphological limitations, the authors should include information regarding the expression o some biomarkers that could help in confirming the diagnosis of USC or exclude other histotypes (ie: p53, RE, RP, HINF1B, ARID1A,  MMR proteins, beta-catenin, etc.).

That some cases included in this series could represent examples of other histotypes is suspected by a relatively low percentage of cases carrying TP53 mutations (70% in present series vs 88% in TCGA).

The authors described a case with a MSH6 pathogenic mutation, that probably is the case with MSI presented in Table 1.  If so, please indicate this association in the text.

Since carcinomas with MMRd are more frequently endometrioid, the authors should include in the main text or in the supplementary material histological images of this case.

Statistical differences among groups (probably excluding the undetermined group)  should be presented in Tables 1 and 2.

The authors should include a multivariate prognosis analysis with stage and TMB as variables.

Reviewer 3 Report

Great paper. Methods were adequately conducted and results are sound. Very well written. 

Author Response

Thank you for your review! We appreciate your time.

Round 2

Reviewer 1 Report

The authors have nicely addressed my all concerns.

Reviewer 2 Report

The authors have improved the MS.